# Progressive Invasion of *Aedes albopictus* in Northern Spain in The Period 2013–2018 and A Possible Association with The Increase in Insect Bites

**DOI:** 10.3390/ijerph17051678

**Published:** 2020-03-04

**Authors:** Fátima Goiri, Mikel Alexander González, Joseba Goikolea, Madalen Oribe, Visitación de Castro, Sarah Delacour, Javier Lucientes, Ione Ortega-Araiztegi, Jesús Felix Barandika, Ana Luisa García-Pérez

**Affiliations:** 1Animal Health Department, NEIKER-Instituto Vasco de Investigación y Desarrollo Agrario, Bizkaia Science and Technology Park 812L, 48160 Derio, Bizkaia, Spain; fgpresmanes@neiker.eus (F.G.); mgdeheredia@neiker.eus (M.A.G.); ioneortegaaraiztegi@gmail.com (I.O.-A.); jbarandika@neiker.eus (J.F.B.); 2Departamento de Salud del Gobierno Vasco, Subdirección de Salud Pública de Gipuzkoa, 20013 Donostia-San Sebastian, Gipuzkoa, Spain; ambien5ss-san@euskadi.eus; 3Unidad de Epidemiologia, Departamento de Salud del Gobierno Vasco, Subdirección de Salud Pública de Bizkaia, 48008 Bilbao, Bizkaia, Spain; m-oribe@euskadi.eus (M.O.); v-castro@euskadi.eus (V.d.C.); 4Animal Health Department, The AgriFood Institute of Aragon (IA2) Faculty of Veterinary Medicine, 50013 Zaragoza, Spain; delacour@unizar.es (S.D.); jlucien@unizar.es (J.L.)

**Keywords:** *Aedes albopictus*, surveillance, ovitraps, human bites, biocides

## Abstract

(1) Background: *Aedes albopictus* has rapidly expanded throughout Europe, becoming a public health concern in the Mediterranean Basin. (2) Methods: Following the detection of *Ae. albopictus* in the southwestern French region of Aquitaine in 2012, an entomological surveillance programme was implemented in the Basque Country (Northern Spain) in 2013. (3) Results: *Ae. albopictus* eggs were first detected in 2014 in a transited parking area in the northeastern sampling point, 22 km away from the nearest French site with recorded presence of tiger mosquito. At this site, eggs were found throughout the study (2014–2018). Other western and southern municipalities became positive in 2017 and 2018. *Ae. albopictus* adults were first captured in 2018 by aspiration of the vegetation in an area where eggs had been detected since 2015, suggesting a progressive establishment of a self-sustained population. Incidence of insect bites in humans was roughly constant over the study period except for a significant increase in 2018 in the Health County where eggs had been detected since 2014. Densities of *Ae. albopictus* eggs in positive areas remained at similar levels over the years. (4) Conclusion: Multiple approaches and standardized methods are necessary to successfully control this vector.

## 1. Introduction 

The Asian tiger mosquito *Aedes albopictus* (Skuse, 1894) is an invasive species that can be easily transported [1,2,3], and therefore, it is currently found in temperate and tropical continents, like Asia, Europe, North and South America, Africa, and a number of locations in the Pacific and Indian Oceans [4]. Introduction of *Ae. albopictus* in Europe was first documented in Albania during the late 1970s. This first description was followed by reports of its presence in Italy in the 1990s, from where it gradually spread to other Mediterranean countries [5]. In addition, its ability to adapt to cold temperatures and overwinter in temperate regions, its plasticity to adapt to different habitats [4], together with globalization and climate change contribute to the successful invasion of *Ae. albopictus*.

*Ae. albopictus* spread is a public health concern since it can act as a vector of some arboviruses, most notably dengue and chikungunya. In fact, several autochthonous outbreaks have been recently reported in Europe, thus confirming local transmission of these diseases in places where *Ae. albopictus* is established [6,7,8]. This situation poses a risk for neighbouring areas where transmission has not yet occurred. In addition, the aggressive biting behaviour of *Ae. albopictus* jeopardises the quality of life of local citizens and can even have a direct impact on regional economy [9]. 

Following the first detection of the Asian tiger mosquito in Catalonia (Spain) in 2004 [10], a surveillance programme funded by the Spanish Ministry of Health started in 2007 at several regions of the Mediterranean Spanish territory [11]. The programme was aimed at detecting as early as possible the presence of tiger mosquito eggs in order to prevent a possible spread and further establishment. Since the increase in the rate of insect bites in people was shown to be a good indicator of the presence and establishment of the *Ae. albopictus* mosquito in Catalonia (Spain) [12], mosquito bite rates can also be included in surveillance programmes. 

Considering the existence of established populations of *Ae. albopictus* in France and the Spanish Mediterranean regions, vehicles travelling through motorway systems would be the foreseeable means for the arrival of the mosquito to northern Spain. This study arises after the detection of eggs of tiger mosquito in 2012 in the French region of Aquitaine (southwestern France) [13], 22 km away from the Spanish Basque Country. This region is limited geographically within southwestern France, and it is located approximately 500 km away from the Spanish Mediterranean areas where tiger mosquito was already fully established [1,10]. The objectives of the current study were (i) to monitor the arrival and spread of *Ae. albopictus* in the period 2013–2018, (ii) to evaluate changes in *Ae. albopictus* egg densities in areas where eggs of tiger mosquito were detected in consecutive years, and (iii) to assess the temporal trends in the rates of insect bites in humans as an indicator of *Ae. albopictus* establishment in the area. 

## 2. Materials and Methods

### 2.1. Study Area 

The Basque Country (northern Spain) (42°78′ N, 02°44′ W), with an extension of ca. 7200 km^2^, is divided into three administrative provinces: Gipuzkoa (NE), Bizkaia (NW), and Araba (S). The climate is warm and temperate. The average annual temperature is 13.4 °C in Gipuzkoa, 13.8 °C in Bizkaia, and 11.5 °C in Araba, and rainfall averages are 1610.6 mm in Gipuzkoa, 1277.7 mm in Bizkaia, and 878.3 mm in Araba [14].

Population of the Basque Country is ca. 2,188,017 inhabitants [15]. Industry is the driving force of the Basque economy and represents 21% of the gross domestic product (GDP) [15]. Due to its geographical location, the region is crossed by a network of highly trafficked motorways structured around the main routes that connect the Basque Country with France, central Spain, and the Mediterranean coast. These roads support around 100,000 tonnes of freight movements per year, being manufactured minerals, building materials, machines, vehicles, and manufactured objects the most common freight, followed by food products [15]. Inbound transport (international and national) represents 49.6% of the movements, whereas outbound transport amounts for the remaining 50.4%. 

Tourism also represents an important part of the economy of the region (5.8–6.1% of GDP). Almost two out of three tourists come from other Spanish regions (37.2% from central Spain, 14.9% from Catalonia, 14.2% from Mediterranean provinces, and 15.6% from other Spanish regions), and the remaining one-third comes from abroad. In fact, due to the geographical proximity, the Spanish Basque Country frequently receives people from the French Basque Country, Les Landes, and French Pyrenees. Only 18% corresponds to internal tourism [15]. 

### 2.2. Surveillance of Ae. albopictus 

Presence of *Ae. albopictus* eggs was investigated using oviposition traps (ovitraps) located in 40 sampling areas on 12 Health Counties (HC) during the period 2013–2018 (Table 1).

The density of the sampling areas varied in number and location along time with an increasing trend over time (Table 1). Ovitraps consisted of a dark container (250 mL) filled with non-chlorinated water and a wooden stick (15 cm long and 2 cm width) submerged inside as an oviposition substrate [16]. Eight to ten ovitraps were placed in each sampling site in wind-protected shaded areas; in many cases, surrounded by vegetation; and near walls or fences. Ovitraps within each sampling site were separated by at least 15–20 m. High or intense traffic of trucks and vehicles was used as selection criteria for sample locations, e.g., motorway service areas, petrol stations, shopping centres, and industrial areas, among others (Table 1).

Entomological surveillance took place between September and November initially (2013–2015) and between July and November afterwards (2016–2018), periods when the highest densities of *Ae. albopictus* mosquitoes have been recorded in Spain [17]. Ovitraps were examined weekly, and the wooden sticks were replaced and transported to the laboratory for examination. The water content of the ovitraps was also visually examined for mosquito larvae, and the water was then replaced using non-chlorinated water. In the laboratory, each labelled wooden stick was studied under a stereoscopic microscope and the eggs with morphology compatible with container breeding *Aedes* mosquitoes were counted. A selection of sticks from positive sites were immersed in small breeders with non-chlorinated water for hatching and further identification of the fourth stage larvae (L4) using taxonomic keys [18]. 

For the collection of adult mosquitoes, two BG Sentinel traps (Biogents, Regensburg, Germany) with attractant lure, two Prokopack aspirators (John Hock, Gainesville, Florida, USA), and five gravid traps (John Hock, Gainesville, Florida, USA) were placed nearby the sites where *Ae. albopictus* eggs were detected. BG Sentinel traps were placed for 24–48 h, and gravid traps were placed for 48 h. An aspirator was used for 20–30 min in vegetation around positive traps. Mosquito identification was also performed using taxonomic keys [18]. 

### 2.3. Control of Ae. albopictus in Positive Sites

Once *Ae. albopictus* eggs were identified in a sampling site, the Department of Public Health of the Basque Country was immediately informed, and local authorities were requested to implement control measures against tiger mosquito in the area. Licensed companies selected by the affected councils carried out biocide treatments (adulticidal and/or larvicidal). Active compounds and dates of application were provided by health officers of the municipalities. 

### 2.4. Monitoring Insect Bite Rates in Humans 

Sampling sites were located in twelve out of 13 HC (Figure 1, Table 1). Insect bite events in each HC were retrospectively compiled from the databases of outpatient health services for the study years (2013–2018) during the period when biting incidence was reported to be higher, i.e., between 1 May and 20 November (weeks 18–48). Patients attended health services after an insect bite when the bite showed signs of irritation or inflammation or when they suffered from an allergic reaction. Bite type was defined by the doctor who attended the patient after bite inspection. Bites included in the study corresponded to Diptera, mainly *Culicidae*, excluding ticks, bees, wasps, bedbugs, or spiders, which are differentially registered. 

### 2.5. Collection of Data and Statistical Analyses

Geographical coordinates of ovitrap location at the 40 sampling sites were recorded using a global positioning system device and were represented in a map using the software QGIS 3.4.12-Madeira (QGIS Geographic Information System, Open Source Geospatial Foundation Project) (Figure 1). Information regarding HC, principal motorways, and roads was obtained from the GeoEuskadi, Infrastructure, and Spatial data platform [19]. 

Annual and weekly mean temperatures (min, max, and mean), humidity, and precipitation were compiled from meteorological stations located in the HC where *Ae. albopictus* eggs were observed during the period 2013–2018 between May and November. Differences observed among years were compared by nonparametric Kruskal–Wallis test. To assess differences among years with regard to the percentage of positive ovitraps, Pearson’s Chi-squared test (categorical variables) was used. *Ae. albopictus* egg density among years and sites were compared by nonparametric Kruskal–Wallis and Wilcoxon two-sample test. Since sampling starting time varied among years, egg density annual means per ovitrap were calculated for the period 1 September–15 November. R software (version 3.4.3, R Foundation for Statistical Computing, Vienna, Austria) was used for statistical analyses. Probability values of *p* < 0.05 were considered significant. The outpatient biting rate per 1000 inhabitants, with confidence intervals at 95%, was assessed by HC and year using SPSS software (version 25.0, IBM SPSS Statistics for Windows, Armonk, NY, USA). Results were considered significant when confidence intervals were not overlapped. Spearman correlation test was used to assess any possible associations between insect bite rates per HC and year with mean annual climatic variables.

## 3. Results 

### 3.1. Surveillance of Ae. albopictus Eggs in The Period 2013–2018

A total of 40 sampling sites were sampled between 2013 and 2018 (Table 1). Eggs were detected in 12 of the 40 areas studied (30%), including most types of selected sites: industrial zones (2/5), shopping centres (2/4), petrol stations (1/5), recycling platforms (1/1), car parks (2/2), or city centres (4/16). No eggs were detected in the six motorway service areas surveyed (0/6). 

All the ovitraps examined in 2013 were negative for *Ae. albopictus* eggs. First detection of *Ae. albopictus* eggs occurred in 2014 in a busy shopping centre parking area in Irun-Behobia (Site 7) close to the French border. This sampling site remained positive until the end of the study. Moreover, during the following four years, tiger mosquito eggs were detected in another three sampling sites within the same municipality: two urban areas which tested positive only in 2015 (site 11) and in 2017 (site 13) and a petrol station (site 10) which remained positive from 2015 to 2018 with a progressive increase in the percentage of positive ovitraps (Figure 2). Between the years 2014 and 2016, tiger mosquito eggs were only identified in a radius of 4 km of the municipality of Irún-Behobia (Gipuzkoa). As the number of municipalities and sampling areas examined increased along the study period, more positive sites were detected. Thus, in 2017, *Ae. albopictus* was detected for the first time in the province of Bizkaia (sites 20 and 23) in areas 100 km away from Irún-Behobia (first detection site) but placed within the main road freight network. Both areas had been sampled the previous year (2016), giving negative results. In 2018, the number of *Ae. albopictus* positive ovitraps increased in both areas (Figure 2) and six new sites became positive for the first time, including the first detection in the province of Araba. Thus, in 2018, *Ae. albopictus* eggs were detected in a caravan parking site (site 34) in Donostia-San Sebastian (Gipuzkoa) very frequently visited by tourists and located 23 km away from Irún-Behobia: four new sites in the province of Bizkaia and an industrial area in Araba (site 6) with important commercial transportation of different goods to nearby provinces (Figure 1). Complete results of the surveillance programme are compiled in Appendix A.

In general, the proportion of positive ovitraps increased over the years (Table 2), the highest percentage being detected in 2018 (3.6%), which represented a significant increase in relation to previous years (X^2^ = 145.27, df = 5, and *p* = 0.0001). In fact, in 2018, only 4 sampling sites accounted for 89% of the positive ovitraps (sites 7, 10, 20, and 23). The number of ovitraps examined and the number of positive traps per sampling site and year are summarised in Appendix A. 

Native *Culicidae* species (*Culex pipiens, Culex hortensis, Culiseta longiareolata, Ochlerotatus caspius, Ochlerotatus detritus*, *Culex* spp., and *Anopheles* spp.) were captured using BG Sentinel traps near site 7 and site 10 in 3–5 samplings carried out between the end of September and the middle of November (2014–2016). Similarly, BG Sentinels were placed in site 20 and *Cx. pipiens, Culex* spp. and *Cs. longiareolata* were identified (2017–2018). Gravid traps were incorporated in 2018 in sites 10, 20, and 25, but no *Ae. albopictus* adults were trapped. In addition, an entomological aspirator was used in 2018 in sites 10, 20, 21, and 22, but only a few specimens of tiger mosquitoes (*N* = 6) were captured at site 10. Data of adult mosquitoes collected and their identification are summarized in Appendix A.

### 3.2. Treatments Applied

Once *Ae. albopictus* eggs were detected in a municipality, the local council was contacted and biocidal treatments were rapidly applied by licensed companies for mosquito and pest control. Each company followed their own strategy in the selection of biocidal compounds and frequency of treatments, and no harmonized procedures were used. Overall, the strategy involved an initial treatment with adulticide (deltamethrin 2.5%) followed by a second treatment at 48 h and a third application 7 days later. Larvicides such as *Bacillus thuringiensis israelensis* spores (sites 7 and 10 for 2016 and 2018), S-metoprene (site 20 for 2017 and 2018), or diflubenzuron (sites 7 and 10 for 2017) were also applied in potential breeding sites surrounding positive sampling locations. Changes in the percentage of positive ovitraps before and after treatments are shown in Figure 2. When the average number of eggs per positive ovitrap was compared over the years, no differences were observed (Site 7, *p* = 0.7238; site 10, *p* = 0.5050; site 20, *p* = 0.1277; and site 23, *p* = 0.0861) (Figure 3).

### 3.3. Progress of Insect Bites in Humans in The Period 2013–2018

The incidence of reported insect bites in the human population was compiled for each HC, where tiger mosquito eggs were detected. A slight increase in biting rates was observed in 2018 in all counties (Figure 4 and Appendix A) compared to years 2016 and 2017, and a significant increase was observed in HC-B (Bidasoa), where tiger mosquito had been detected since 2014 (incidence in 2018: 6.1 bites/1000 inhabitants, CI 95%: 5.6–6.5 *vs*. incidence 2016–2017: 4.1–4.4 bites/1000 inhabitants, CI 95%: 3.5–4.9). In the other HCs, bite rates remained stable throughout the entire study period. Also to be noted is the higher biting rates recorded throughout the whole study period in HC-BG (Barrualde-Galdakao) (incidence in 2016–2018: 6.5–8.1 bites/1000 inhabitants, CI 95%: 6.3–8.3) compared to all other HCs (Figure 4) even though presence of tiger mosquito in this HC was not detected until 2017.

Overall, taking into account annual mean temperatures, humidity, and total precipitation, a positive correlation was found between the rate of insect bites per 1000 inhabitants and the humidity (r = 0.41, *p* = 0.0129). Moreover, a negative significant association between bite rates and precipitation (r = −0.34, *p* = 0.0415) was observed. No significant correlation was found between bite rates and the different temperatures (min, max, and mean) studied. A significantly higher humidity with respect to previous years (*p* < 0.05) was observed in 2018 in sampling site 10 (HC-B, Bidasoa). Regarding temperatures, no statistical differences were observed among years in the HC where *Ae. albopictus* eggs were detected. 

## 4. Discussion

This surveillance programme has provided valuable data to better understand the invasion process of *Ae. albopictus* in the Basque Country. The recommended tools in epidemiological surveillance systems, consisting of the periodic examination of ovitraps along with the use of BG Sentinel and other devices for the capture of adult mosquitoes [16], were used, and the overall findings suggested that *Ae. albopictus* can be easily introduced in any area with abundant freight traffic. However, due to limitation of resources, the programme started with a smaller number of sampling sites that increased to cover the three Basque provinces when, in 2016, other institutions, like local councils and health services, became engaged. Consequently, in those places where the monitoring programme was not initially implemented, only a partial picture was obtained, thus hampering mosquito tiger expansion interpretation.

*Ae. albopictus* eggs were detected the second year of the programme, first detections being restricted to sites with dense vehicle traffic within a municipality near the French border. The following years, mosquito eggs were spotted in sites located 100 km away from its original detection site, so that introduction of *Ae. albopictus* from regions other than France like the Spanish Mediterranean regions where tiger mosquito is fully established [11] cannot be ruled out. In this case, road traffic would have contributed to the spread of this invasive species as previously reported [1]. On the other hand, the continued and increased presence of eggs year after year in the same locations near the French border indicated not only possible new arrivals through transport but also local reproduction and colonization of *Ae. albopictus* [20] as reported in other regions. In contrast, the sporadic detection in other sites would be indicative of a recent introduction. However, its survival and establishment success would be driven mainly by climatic conditions and land use, with urban and peri-urban areas providing abundant suitable sites for egg laying [4,21]. 

Climatic conditions are key in the establishment of tiger mosquito, and different predictive models have been projected for Europe in the past years [22,23,24]. Annual mean temperature (above 10 °C) and particularly mean temperature in January (above 0°) are the two crucial variables for the expansion of this species. January mean temperature conditions the survival of the eggs in winter and the ability of the population to overwinter [21]. According to Roiz et al. [23], rainfall would not be a limiting factor for mosquito establishment if mean annual precipitation is above 500 mm. In the Basque Country, neither rainfall nor temperatures of January would be critical limitations [14]. Regarding temperature, *Ae. albopictus* growth is dependent on external temperature and is likely to be slower in temperate regions with cool summers [25] as is the case in the Basque Country (or at least the northern parts of the Basque Country). This temperature threshold might have acted as a barrier to delay the establishment of the *Ae. albopictus* populations in the Basque Country and could explain the failure to detect adults during the first five years after the first detection of eggs. Another alternative is that *Ae. albopictus* populations grew so slowly over those years that their growth was undetectable. This is in agreement with previous studies that reported that *Ae. albopictus* population requires a few years to be established, as seen in Switzerland [26], and to start causing problems in human welfare. The slight increase in biting rates observed in 2018 in all HCs where tiger mosquito eggs were present might also be an indicator to support that hypothesis. However, it is difficult to know if *Ae. albopictus* could have been involved in the increase in bites in 2018 since an increase in the population of native *Culicidae* has also been reported in several European countries due to the high temperatures in summer 2018 [27]. However, temperatures in the Basque region were not significantly higher in 2018 in comparison with previous years, and only humidity showed a significant increase in 2018. Still, the increase in reported biting rates observed in HC-B (Bidasoa), where tiger mosquito was present since 2014, was significant, suggesting that the presence of the mosquito could have started having an impact on the well-being of the population. However, insect bite rates in the Basque population are far lower than rates observed in other Spanish regions (Catalonia) at the time *Ae. albopictus* was first observed (16 consultations/1000 inhabitants in 2004) [12]. Therefore, surveillance of biting rates in the Basque Country in future years will be very useful to monitor the presence and abundance of the *Ae. albopictus* population and to evaluate the efficacy of the control actions implemented to minimize its effects on citizens welfare.

The control of invasive species is challenging, and in most cases, its result is barely effective or short-effective [28]. However, some success in the reduction of *Ae. albopictus* populations was reported in Spain and Switzerland [26,29]. Once tiger mosquito is established in an area, a combination of multiple control measures need to be applied [20] to restrain spread and to keep the population at low levels. To do so, public institutions and community interventions play a very important role [13,26,30]. In fact, in 2018, the public health authorities in the Basque Country started the diffusion of leaflets containing information about the biology of the mosquito and the importance of recognizing and removing breeding sites suitable for *Ae. albopictus* in private properties of the affected municipalities. In this study, the treatments applied at the positive sites were irregular and intermittent, different operators used different active products and doses, and surfaces treated were unknown. This could be in part consequence of the limited experience of local control companies in the management of this invasive species and the restrictions in treatment budgets. Lack of harmonized control protocols hampered results comparison and prevented the implementation of common action plans at the affected municipalities. Future actions implemented in the upcoming years to fight against this vector species should overcome this situation. 

The density of eggs in ovitraps gives an indication of mosquito-biting females’ densities, and the mean number of eggs is used in surveillance and control programmes [26,29]. Interestingly, the median eggs per positive trap remained at stable levels in the sampling sites where treatments were applied, with averages below 60 eggs/positive ovitrap in the six years of the programme. Similar results were found in Switzerland in the first five years of the surveillance programme (2000–2004) when <50 eggs per positive ovitrap were reported [26]. The results presented here and the *Ae. albopictus* trends reported in other studies suggest that invasion will progress throughout Europe and highlights the potential impact of climate change on *Ae. albopictus* populations [22]. 

It is been over four decades now since the *Ae. albopictus* colonization process started in Europe. Since then, molecular genetic studies have demonstrated the invasion progress in different countries from the original sources in Albania, northern and central Italy [31]. Similar studies would be needed to trace back the introduction of tiger mosquito into the Basque Country and to identify the sources.

## 5. Conclusions

This comprehensive entomological study highlighted the risk of a rapid spread of this alien species in non-native areas. Despite surveillance programmes and the application of control measures, it seems difficult to stop the introduction and expansion of *Ae. albopictus* in the Basque Country region due to the high flow of vehicle trade. To monitor spread and to achieve early detection, the surveillance programme is ongoing and new sampling locations will be included to cover new areas. Also, insect biting surveys should be included as part of *Ae. albopictus* surveillance programmes, considering also human landing catches [32]. On the other hand, future control measures in positive sites should follow standardized protocols. Finally, local administrations in collaboration with research centres should promote passive surveillance by involving the community (the so-called citizen science) as a cheap advantageous surveillance alternative [13].

## Figures and Tables

**Figure 1 ijerph-17-01678-f001:**
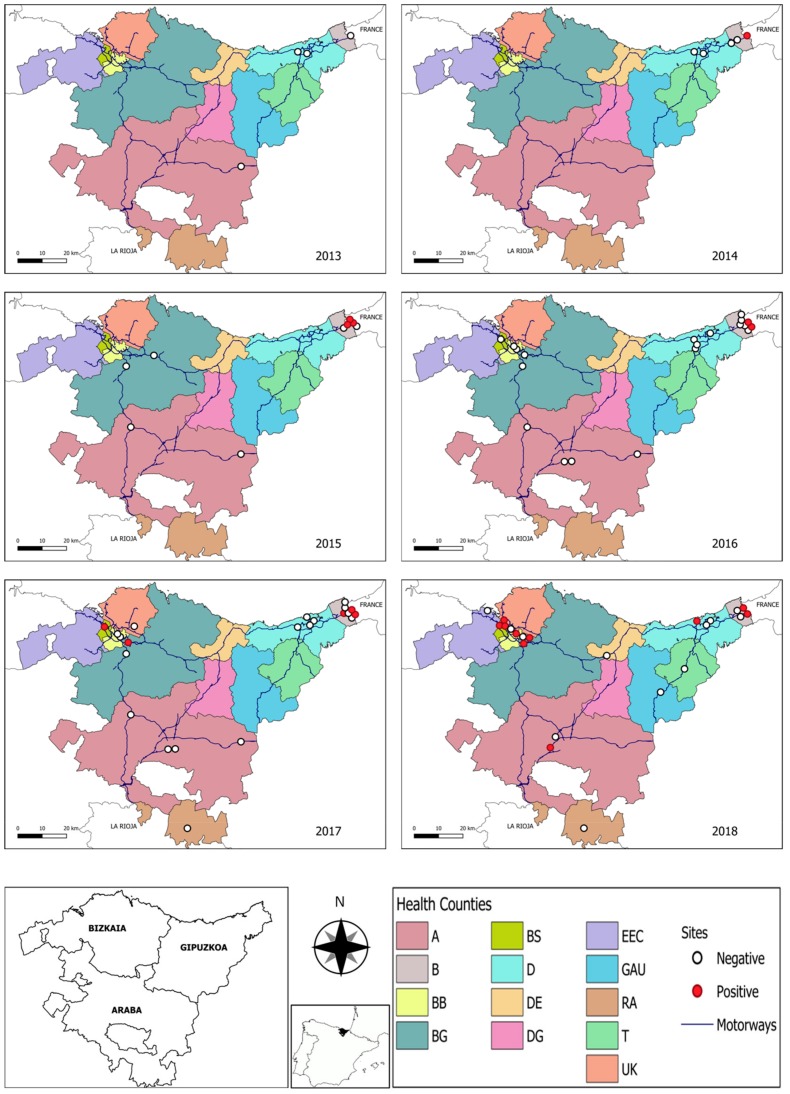
Maps of the Basque Country indicating the sites where ovitraps were placed in each year along the surveillance programme (2013–2018) (Source: Own elaboration based on GeoEuskadi data). Open circles denote negative sites, whereas presence of *Ae. albopictus* eggs in at least one ovitrap is indicated by red circles.

**Figure 2 ijerph-17-01678-f002:**
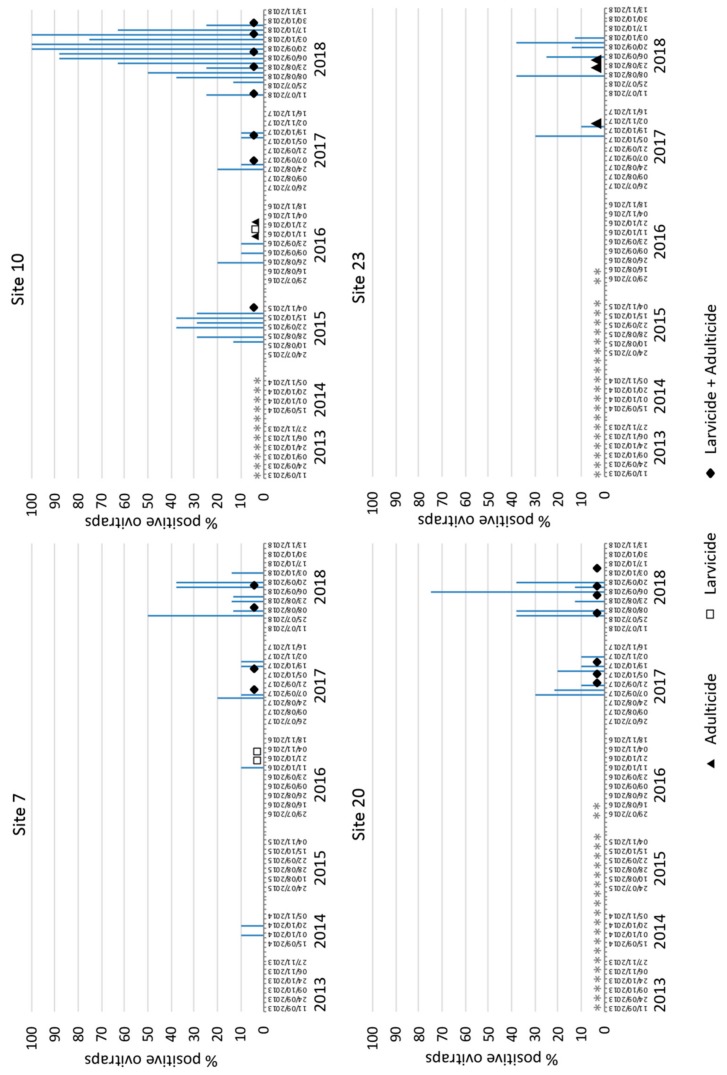
Percentages of positive ovitraps in the 4 sites where *Ae. albopictus* eggs were observed throughout the study and timing of treatments applied: Asterisks indicate dates when sampling was not performed.

**Figure 3 ijerph-17-01678-f003:**
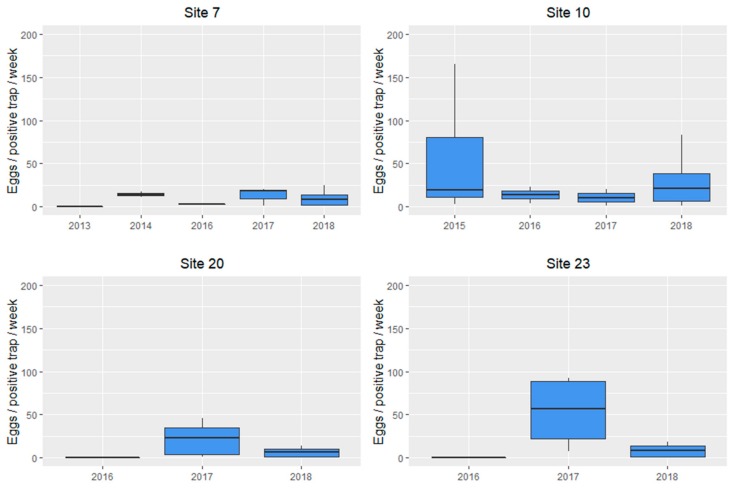
Boxplot showing the number of eggs per positive ovitrap over the years in the 4 sampling sites where *Ae.albopictus* eggs were observed during consecutive years: Boxplot graphs represent the median burden, the lower and upper quartiles (boxes), and minimum and maximum values (whiskers).

**Figure 4 ijerph-17-01678-f004:**
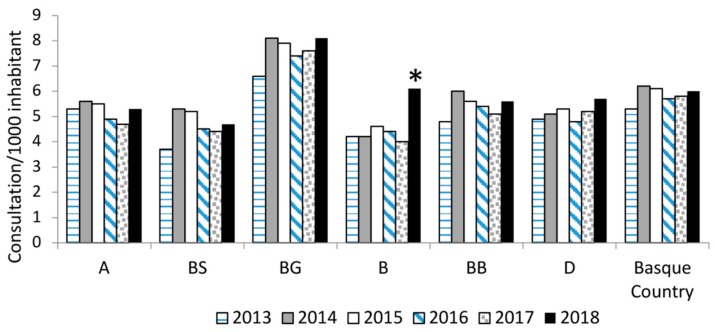
Insect bite consultations per 1000 inhabitants in the Health Counties where *Ae. albopictus* eggs were detected for the period 2013–2018 (A = Araba; BS = Barakaldo-Sestao; BG = Barrualde-Galdakao; B = Bidasoa; BB = Bilbao Basurto; D = Donostiadea). Significant increases are indicated by asterisks.

**Table 1 ijerph-17-01678-t001:** Sampling sites description and year of detection of *Ae. albopictus* eggs.

HC ^1^	Site	Site Description	Coordinates 9/UTM 30)	Sampling Year	Years of Detection
A	1	Petrol station	X:555518 Y:4745944	2013, 2015–2017	
	2	Service area	X:510280 Y:4757046	2015–2017	
	3	City centre	X:526669 Y:4742969	2016–2017	
	4	City centre	X:525569 Y:4742846	2016–2017	
	5	Airport	X:521831 Y:4747911	2018	
	6	Industrial area	X:519711 Y:4743493	2018	2018
B	7	Parking	X:600521 Y:4799707	2013–2018	2014–2018
	8	Shopping centre	X:594423 Y:4797251	2014	
	9	Petrol station	X:601109 Y:4798864	2015	
	10	Petrol station	X:600172 Y:4799291	2015–2018	2015–2018
	11	City centre	X:597750 Y:4799166	2015–2017	2015
	12	City centre	X:598244 Y:4798572	2015–2018	
	13	City centre	X:597039 Y:4799326	2016–2018	2017
	14	Petrol station	X:597768 Y:4801111	2016–2017	
	15	City centre	X:597590 Y:4803050	2016–2017	
BB	16	City centre	X:505938 Y:4789059	2016–2018	2018
	17	City centre	X:504805 Y:4790265	2016–2018	
BG	18	Service area	X:508383 Y:4782104	2015–2017	
	19	Service area	X:519772 Y:4786539	2015	
	20	Industrial area	X:509199 Y:4786765	2016–2018	2017–2018
	21	Recycling centre	X:508893 Y:4786442	2018	2018
	22	City centre	X:508622 Y:4786697	2018	
BS	23	Shopping centre	X:499449 Y:4793195	2016–2018	2017–2018
	24	Shopping centre	X:499764 Y:4792537	2018	2018
	25	City centre	X:499907 Y:4793331	2018	2018
D	26	Shopping centre	X:585497 Y:4795662	2016–2018	
	27	Service area	X:578823 Y:4793078	2013–2017	
	28	Service area	X:582588 Y:4792374	2013–2014	
	29	Service area	X:593455 Y:4796905	2014	
	30	Industrial area	X:579356 Y:4790133	2016	
	31	Petrol station	X:579405 Y:4789368	2016	
	32	City centre	X:582592 Y:4797197	2017	
	33	City centre	X:583839 Y:4793884	2017–2018	
	34	Parking	X:579931 Y:4795501	2018	2018
DE	35	City centre	X:542890 Y:4781322	2018	
EEC	36	Industrial area	X:493897 Y:4799791	2018	
GAU	37	City centre	X:565017 Y:4766285	2018	
RA	38	City centre	X:533517 Y:4710345	2017–2018	
T	39	City centre	X:574834 Y:4775857	2018	
UK	40	Industrial area	X:511692 Y:4793419	2017	

^1^ HC, Health County (A: Araba; B: Bidasoa; BB: Bilbao-Basurto; BG: Barrualde-Galdakao; D: Donostialdea; DE: Debabarrena; EEC: Eskerraldea; GAU: Goierri Alto Urola; RA: Rioja Alavesa; T: Tolosaldea; UK: Uribe Kosta).

**Table 2 ijerph-17-01678-t002:** Summary of sampling details and results on the presence of *Aedes albopictus* eggs in the ovitraps over the study years and locations.

Year	N ^1^ Sampling Sites	N Sampled Municipalities	N Ovitraps Examined	Positive Ovitraps N (%)	Positive Sampling Sites ^2^	N Municipalities with *A. A.^3^* (HC) ^4^
2013	4	4	404	0 (0.0)	0	0 (na ^5^)
2014	5	5	448	2 (0.4)	7	1 (B)
2015	9	5	810	15 (1.9)	10, 11	1 (B)
2016	20	11	2801	5 (0.2)	7, 10	1 (B)
2017	22	13	2531	25 (1.0)	7, 10, 13, 20, 23	3 (B, BG, BS)
2018	23	11	3436	123 (3.6)	6, 7, 10, 16, 20, 21, 23, 24, 25, 34	6 (A, B, BB, BG, BS, D)

^1^ N, number; ^2^ Sampling sites as described in Table 1; ^3^ A.A., *Aedes albopictus*; ^4^ HC, Health County (A, Araba; B, Bidasoa; BB, Bilbao-Basurto; BG, Barrualde-Galdakao; BS, Barakaldo-Sestao; D, Donostiadea); ^5^ na, not applicable.

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
