# Peer review of "Progressive Invasion of Aedes albopictus in Northern Spain in The Period 2013–2018 and A Possible Association with The Increase in Insect Bites"

_ijerph, 2020, doi:10.3390/ijerph17051678_

Round 1

Reviewer 1 Report

Dear authors,

This manuscript deals with the study of presence and spread of Ae. albopictus in the northern Spain collecting data from ovitraps, BG-sentinel traps and reported mosquito bites in humans in areas where control methods were applied. The findings are interesting since they provide evidence for the presence and spread of tiger mosquito from 2013 to 2018 in the studied area.

Herein, you will find some comments that may benefit the manuscript.

General comments:

As a drawback of this study seems to be the fact that there was not an established permanent network of oviposition traps in all sampling sites throughout the whole studied period, in order to provide robust conclusions concerning the presence and spread of Ae. albopictus in northern Spain from 2013 to 2018 and allow sound comparisons of mosquito population between areas and years. Please comment.

I propose to remove “…its impact on human bites” from the title of the manuscript, since the correlation of Ae. albopictus population with the reported mosquito bites was not thoroughly studied using statistics and moreover the reported bites may have included bites from other mosquito species. Hence, I suggest removing the third objective of this study in the introduction concerning the assessment of tiger mosquito bites as an indicator of Ae. albopictus establishment.

Do you have any meteorological data (temperature, precipitation) from the studied sampling sites or Health counties that may help in the interpretation of the results concerning Ae. albopictus population presence and bites?

In this perspective, any available data from other mosquito control programs implemented in the studied areas prior to or after Ae. albopictus detection should be addressed in the manuscript since they may have affected its presence in the treated areas.

Specific comments:

Line 85: It’ s better to refer to Health counties, according to table 1, instead of localities.

Line 95: Please clarify the distance between ovitraps.

Lines 105-107: What was the hatching rate of the larvae? Do you have any data? Did you find any other container breeding mosquitoes in the ovitraps?

Lines 108-111: Please clarify if the adult mosquito traps (BG-sentinel and gravid traps) were monitored in each sampling site and each sampling date after Ae. albopictus detection. How do you explain the few specimens captured in BG-sentinel traps at only one site in one year (year) compared with the considerable % of positive sampling sites over the tested period, especially in 2017 (5 positive from 22 in total) and 2018 (10  positive from 23 in total)?

Line 189: In 2018, although almost half of the studied sampling sites were positive for Ae. albopictus, only a very low percentage of the ovitraps was positive. Can you explain this finding?

Line 219 (Figure 3): In Y axis it should be eggs/per positive trap/week.

Lines 284-286: I suggest removing these lines, because in site 23 there was an increase of egg collection from 2016 to 2017, and in site 10 there was an increase of Ae. albopictus eggs from 2017 to 2018 despite the intensive control measures taken that year. Can you justify the considerably high Ae. albopictus egg collections in 2018 at site 10 despite the repeated larvicidal and adulticidal applications?

Line 229: In the discussion you could evaluate the % of positive ovitraps per year (which seems to be very low), the number of eggs per week per positive ovitrap and the biting rate found in the studied areas, in comparison with literature data from relevant field studies in other regions of Spain and other European countries.

Author Response

Reviewer 1

Dear authors,

This manuscript deals with the study of presence and spread of Ae. albopictus in the northern Spain collecting data from ovitraps, BG-sentinel traps and reported mosquito bites in humans in areas where control methods were applied. The findings are interesting since they provide evidence for the presence and spread of tiger mosquito from 2013 to 2018 in the studied area.

Herein, you will find some comments that may benefit the manuscript.

General comments:

As a drawback of this study seems to be the fact that there was not an established permanent network of oviposition traps in all sampling sites throughout the whole studied period, in order to provide robust conclusions concerning the presence and spread of Ae. albopictus in northern Spain from 2013 to 2018 and allow sound comparisons of mosquito population between areas and years. Please comment.

AU: Yes indeed, at the beginning we started with only 4 sampling sites that gradually increased over the next two years with 5 and 9 sites respectively. Until then, NEIKER did all the samplings, and the resources were limited. Not until 2016 that the programme was expanded to 20 sampling sites with the help and collaboration of the local councils of the capitals of the three Basque provinces and the personnel of Public Health. The strategy at the beginning was when a municipality was positive, the following year the sampling sites in that municipality was then incremented in order to survey the possible expansion of this species. For the rest of sampling sites, at the end of each sampling season it was reviewed whether it was interesting to maintain those ovitraps or moving them to new areas that presented a higher probability to discover eggs of tiger mosquito.

I propose to remove “…its impact on human bites” from the title of the manuscript, since the correlation of Ae. albopictus population with the reported mosquito bites was not thoroughly studied using statistics and moreover the reported bites may have included bites from other mosquito species. Hence, I suggest removing the third objective of this study in the introduction concerning the assessment of tiger mosquito bites as an indicator of Ae. albopictus establishment.

AU: We found of interest to include in the title the part on the surveillance of insect bites on the population seeing as there are very few studies that include these reports in the surveillance programmes of Ae. albopictus. As stated, the bites could have been caused by other insects different than tiger mosquito; we therefore have changed the title and the third objective (lines 67-69).

Do you have any meteorological data (temperature, precipitation) from the studied sampling sites or Health counties that may help in the interpretation of the results concerning Ae. albopictus population presence and bites?

AU: We have compiled meteorological data from May to October, as well as the annual data from meteorological stations located near the positive sampling sites. Correlation analyses have been performed between the rate of mosquito bites per 1,000 inhabitants with the mean temperatures (min, max and mean), humidity and precipitation (lines 160-162). In the new version of the manuscript these new results have been included (lines 262-266). We have found a significant and positive correlation between the rate of insect bites and humidity (r=0.41, P=0.0129), and a negative significant association with precipitation (r= -0.34, P=0.0415).

In this perspective, any available data from other mosquito control programs implemented in the studied areas prior to or after Ae. albopictus detection should be addressed in the manuscript since they may have affected its presence in the treated areas.

AU: In the studied area, there are usually no complaints from the population for mosquito bites, at least until now, in comparison with other areas in Spain. Therefore, no control programme has been implemented against culicids or other insects in this area.

Specific comments:

Line 85: It’s better to refer to Health counties, according to table 1, instead of localities.

AU: Change done (line 94)

Line 95: Please clarify the distance between ovitraps.

AU: In each sampling site 8 to 10 ovitraps were placed and the distance between them varied (15-20 meters) depending on the range of the area. In the manuscript, a sentence specifying this information has been added (lines 105-106).

Lines 105-107: What was the hatching rate of the larvae? Do you have any data? Did you find any other container breeding mosquitoes in the ovitraps?

AU: The first two years of the programme the sticks with the eggs were sent to the Reference Laboratory of Invasive Mosquitoes (University of Zaragoza), and the hatching rate and identification of the instar 4 larvae were carried out in that laboratory. Afterwards, the hatching and identification of the eggs were performed in NEIKER. The hatching rates and the identification were performed in the first positive traps in all positive sites. As the study was not carried out systematically through all the samplings, the results obtained have not been included in the present manuscript. According to this, and to avoid misunderstandings, the corresponding sentence in Material & Methods has been changes (lines 116-118). Taking into account our experience in the lab, the hatching rate is around 30% (range 0-100%).

As for any other breeding mosquito species found in the traps, occasionally 4th instar larvae of Culex spp. and Culiseta spp. were found.

Lines 108-111: Please clarify if the adult mosquito traps (BG-sentinel and gravid traps) were monitored in each sampling site and each sampling date after Ae. albopictus detection. How do you explain the few specimens captured in BG-sentinel traps at only one site in one year (year) compared with the considerable % of positive sampling sites over the tested period, especially in 2017 (5 positive from 22 in total) and 2018 (10  positive from 23 in total)?

AU: According to findings obtained in the field, BG-Sentinel traps were placed near the sampling sites: site 7 (2014-2016); site 10 (2015-2016) and site 20 (2017-2018). In 2018, an entomological aspirator was used in sites 10, 16, 20, 21 and 22. Also gravid traps were used in 2018 in sites 10, 20 and 25. Ae.albopictus was trapped only in site 10 with the aspirator in September 2018 (6 adults). In the new version of the manuscript a summary table has been included detailing the adult samplings performed, as well as, the obtained results (Table S3). This table has been included as supplementary material.

We cannot explain these low rates of Ae.albopictus presence; it may be due to the fact that the population had not been yet fully established in the period between 2014 and 2017, thus the presence of adults was not high enough to be trapped easily.

Line 189: In 2018, although almost half of the studied sampling sites were positive for Ae. albopictus, only a very low percentage of the ovitraps was positive. Can you explain this finding?

AU: Yes, this is due to the fact that from the 10 positive sampling sites recorded in 2018, only 4 of them added 89% of the positive ovitraps (sites 7, 10, 20 and 23), whereas for the rest of the 6 sampling sites the number of positive ovitraps was very low (sites 16, 24, 41 with only 1 positive trap; site 6 with 2 positive traps; site 25 with 3 positive traps; site 21 with 5 positive traps). In the new version of the manuscript a comment has been added about this information (lines 197-199). Also Table S2 has been included as supplementary material, including a summary of the number of ovitraps examined and number of positive ovitraps, per sampling site and year, for a better understanding of previous comments.

Line 219 (Figure 3): In Y axis it should be eggs/per positive trap/week.

AU: Figure 3 has been modified according to the reviewer’s comment.

Lines 284-286: I suggest removing these lines, because in site 23 there was an increase of egg collection from 2016 to 2017, and in site 10 there was an increase of Ae. albopictus eggs from 2017 to 2018 despite the intensive control measures taken that year. Can you justify the considerably high Ae. albopictus egg collections in 2018 at site 10 despite the repeated larvicidal and adulticidal applications?

AU: Despite the increase in positive traps in the studied sampling sites, the mean of eggs per year per positive ovitrap has not experienced a significant increase over the years, we found this data interesting and we attributed it to the treatments applied, but it could be due to other factors. We have removed these lines. Regarding the explanation on why, despite the treatments applied, the number of positive traps has increased in site 10, which is located near the French border; this may be due to the continued entry of Aedes during the summer season, and the lack of effectiveness of the applied adulticides, which did not reach adult mosquitoes hiding in the surrounding vegetation. Moreover, the companies in the sector, in general, are not familiar with the tiger mosquito, and especially with the identification of the breeding sites and with the larvicide treatment guidelines. Lastly, the problem may also lie in the fact that the municipality involved has not implemented any comprehensive plan for the entire municipality of Irún (where sites 7 and 10 are located), and at the moment only the positive areas are treated, even though Ae.albopictus might be more widespread than what was seen in this study.

Line 229: In the discussion you could evaluate the % of positive ovitraps per year (which seems to be very low), the number of eggs per week per positive ovitrap and the biting rate found in the studied areas, in comparison with literature data from relevant field studies in other regions of Spain and other European countries.

AU: Thank you for the suggestion. In the new version of the manuscript a new paragraph has been added comparing our findings with other similar studies (lines 318-320; lines 340-344).

Thank you for your review and for all the comments, which have helped improve the paper.

Reviewer 2 Report

Sorry but this paper is not providing any news of large interest. A short note for the finding of Aedes albopictus in the Basque area would be more appropriate.

Author Response

Reviewer 2

Sorry but this paper is not providing any news of large interest. A short note for the finding of Aedes albopictus in the Basque area would be more appropriate.

AU: Thank you for the comment. In our opinion, we thought it could be of interest to describe the progression of the tiger mosquito since its introduction in the Basque Country in 2014, and evaluate retrospectively the insect bites in humans in the Health Counties where tiger mosquito eggs were detected. We think there is not much bibliography available on this topic, and we thought that this study provided information on the time needed for the mosquito to be fully established.

Reviewer 3 Report

GENERAL COMMENTS:

The manuscript “Progressive invasion of Aedes albopictus in Northern Spain in the period 2013-2018 and its impact on human bites” is a description of a relatively recent surveillance program in conjunction with detection results, analysis of cases of human bites, and an evaluation of control measures. The manuscript is readable, although some editing will be required to correct a number of grammatical errors. For example, the plural of ‘mosquito’ is used inconsistently (‘mosquitoes’ vs ‘mosquitos’).

The greatest value of this paper is the description of the surveillance program, and the important baseline data it provides on the early stages of Aedes albopictus invasion into Northern Spain. It is difficult to predict or control the expansion of this exotic pest without such baseline data. I do suspect that the legend in Figure 1 has switched positive ovitraps and negative ovitraps, as it appears that ovitraps were initially positive but then became negative after a few years. The data make much more sense the other way around.

The linkage of bite reports to mosquito numbers is an intriguing avenue. I have misgivings as to its usefulness as presented, given that it is unclear how many of these bites are from Aedes albopictus, or even necessarily from mosquitoes.

I am most concerned about the soundness of the evaluation of mosquito control with insecticides. While I appreciate the fact that mosquito control was carried out by a variety of different operators who used different methods, the absence of any specific description of the formulation used, the amount applied, and the locations where control was applied, means that treatment likely varied in thoroughness and quality, so there is no way to draw any conclusions between treatment and Aedes albopictus collections. Furthermore, it is not clear what the negative control was  in this evaluation.

Despite the manuscript’s limitations, it is still useful from an Aedes albopictus surveillance point of view. The paper can be strengthened by adding more specific recommendations for an Aedes albopictus surveillance and control program. I would recommend making the changes described below.

SPECIFIC COMMENTS:

ABSTRACT

Lines 27-29: “Biocide treatments were applied and kept densities of Ae. albopictus eggs remained at similar levels over the years”. First, remove “remained”, it is redundant. Second, the results do not support this conclusion. See later comments for further detail.

INTRODUCTION:

Line 38. “public health concern” is not capitalized.

MATERIALS AND METHODS:

Line 105. What is meant by “eggs with morphology compatible for Ae. albopictus”? The language makes it sound like eggs were identified to species-level. I would suggest using “eggs with morphology compatible for container-breeding Aedes mosquitoes” for clarity.

Lines 113-118. I appreciate the fact that mosquito control was carried out by a variety of independent operators who use different methods, but without knowledge of a) what formulation insecticide(s) were used, b) what quantity was used, and c) where applications were made, there is no way to determine the thoroughness or quality of the insecticide applications. Therefore, there is no way to tell if the treatments had an effect or not (maybe some treatments were very light, maybe some used an inappropriate chemical, maybe some did adulticide and some did not). There is no way to tell if mosquito control has an effect on Aedes albopictus numbers or not based on this information.

Lines 120-126. More detail is needed on these bite cases. Does the patient bring the insect or a description of the insect when reporting the bite? How would the doctor know if it’s a bedbug, simuliid, chemical burn, environmental irritatant, or other medical condition? Oftentimes people report being bitten by arthropods when there is no evidence of arthropods. Without more detail (or perhaps evidence that the vast majority of bite reports are Aedes albopictus bites, at which point you can use total bite reports as a proxy for Aedes albopictus bites), you can’t really link changes in bite reports to presence of Aedes albopictus in an area.

Lines 137-139. 95% confidence intervals were generated using SPSS, but was there any sort of statistical test done? That should be described here.

Figure 1. It would be helpful if the regions were labelled. It is not easy to cross-reference the figure with the text.

RESULTS

Line 151. “All the ovitraps examined in 2013 were negative for Ae. albopictus eggs”. According to Figure 1 and the Figure 1 legend, all of the ovitraps were positive. Could it be the legend has switched which ovitraps are positive and which are negative?

Line 200. I assume you mean Bacillus thuringiensis israelensis? There are multiple strains of Bt, and israelensis is the most likely to be used against Ae. albopictus larvae in the habitats described in this manuscript.

Line 201. Diflubenzuron should not be capitalised.

Lines 203-205. The method by which treatment was statistically assessed needs to be described more in Methods. What was the control?

DISCUSSION

Lines 230-269. This paragraph is too long. It should be broken down into three to five paragraphs.

Lines 243-244. I’m not sure what is meant by “short time detection”. Rapid detection?

Line 246. Replace “egg lying” with “egg laying”

Lines 256-258. “This temperature threshold…could explain the absence of adults during the first five years after the first detection of eggs”. I recommend replacing “absence of adults” with “failure to detect adults”. Clearly adults were not absent from the area, as they were laying eggs.

Lines 262-264. “However, it is difficult to confirm that Ae. albopictus was the solely reason for the increase in bites in 2018 since an increase in the population of native Culicidae has also been reported in several European countries due to the high temperatures in summer”. I think this really sheds doubt on the ability of increased biting reports, even in Bidasoa, to incriminate Ae. albopictus as the cause of the bites. Perhaps one message drawn from this manuscript should be the inclusion of biting surveys as part of Aedes albopictus surveillance, where mosquitoes are identified.  Also, change "solely" to "sole".

Lines 284-286. “In spite of these limitations, the average number of eggs per positive ovitraps and year did not increase over time in the sites with Ae. albopictus established populations (sites 7, 10, 20 and 23), suggesting that the applied biocide treatments helped to maintain low densities of mosquitoes”.   The results do not support this conclusion. An alternative hyposthesis is that Aedes albopictus populations grew so slowly over those years that their growth was undetectable. In order to show that insecticide treatment kept populations low, there would need to be consistency in insecticide applications (the type, amount and location of application would need to be stated, so the reader would know that a similar amount of control was applied in all areas), and there should be a control area where Aedes albopictus was untreated, or at least a comparison made with an area where Aedes albopictus were not treated, to show that Aedes albopictus populations will increase if untreated.

CONCLUSIONS

As stated before, I would consider ways to improve the correlation between Aedes albopictus numbers and reports of humans being bitten. One suggestion is to have human landing catches as part of the surveillance program, in order to measure biting rates accurately.

Author Response

Reviewer 3

GENERAL COMMENTS:

The manuscript “Progressive invasion of Aedes albopictus in Northern Spain in the period 2013-2018 and its impact on human bites” is a description of a relatively recent surveillance program in conjunction with detection results, analysis of cases of human bites, and an evaluation of control measures. The manuscript is readable, although some editing will be required to correct a number of grammatical errors. For example, the plural of ‘mosquito’ is used inconsistently (‘mosquitoes’ vs ‘mosquitos’).

AU: We apologize for the grammatical errors made. We have done our best to correct them in the new version of the manuscript.

The greatest value of this paper is the description of the surveillance program, and the important baseline data it provides on the early stages of Aedes albopictus invasion into Northern Spain. It is difficult to predict or control the expansion of this exotic pest without such baseline data. I do suspect that the legend in Figure 1 has switched positive ovitraps and negative ovitraps, as it appears that ovitraps were initially positive but then became negative after a few years. The data make much more sense the other way around.

AU: Yes, the reviewer is right. The legend is incorrect, red colour represents the positive sites, whereas white represents the negative sites. In the new version of the manuscript this mistake has been amended.

The linkage of bite reports to mosquito numbers is an intriguing avenue. I have misgivings as to its usefulness as presented, given that it is unclear how many of these bites are from Aedes albopictus, or even necessarily from mosquitoes.

AU: It is true that bite reports are not specific of Ae.albopictus, as all bites of Diptera are included, so many of these bites are expected to be mosquitoes Culicidae. Despite this fact, reported biting rates are considered a good indicator to assess the presence of Aedes albopictus, in an indirect way. Monitoring bites enable to establish a baseline of biting rate when there is no evidence of presence of Ae. albopictus and changes in these rates may indicate the presence of Ae. albopictus. In fact, in Catalonia (Spain), in 2004, the presence of tiger mosquito was detected due to the alarming increase in the rate of insect bites in humans (Gimenez et al., 2007).

I am most concerned about the soundness of the evaluation of mosquito control with insecticides. While I appreciate the fact that mosquito control was carried out by a variety of different operators who used different methods, the absence of any specific description of the formulation used, the amount applied, and the locations where control was applied, means that treatment likely varied in thoroughness and quality, so there is no way to draw any conclusions between treatment and Aedes albopictus collections. Furthermore, it is not clear what the negative control was in this evaluation.

AU: The reviewer is right in all he/she says. The only information we were provided with, by the local authorities, were the type of active compound used and the day the treatment was applied. We had tried to explain it in the text and to represent it in Figure 2, that the results were not satisfactory. Now, we have tried not to make claims regarding the effectiveness of the treatments.

No negative control was used, since as soon as Aedes eggs were observed in a sampling site, the corresponding local authorities were informed and control methods were applied.

Despite the manuscript’s limitations, it is still useful from an Aedes albopictus surveillance point of view. The paper can be strengthened by adding more specific recommendations for an Aedes albopictus surveillance and control program. I would recommend making the changes described below.

SPECIFIC COMMENTS:

ABSTRACT

Lines 27-29: “Biocide treatments were applied and kept densities of Ae. albopictus eggs remained at similar levels over the years”. First, remove “remained”, it is redundant. Second, the results do not support this conclusion. See later comments for further detail.

AU: Thank you. The sentence has been modified according to the comments and suggestions of the reviewer.

INTRODUCTION:

Line 38. “public health concern” is not capitalized.

AU: Change done

MATERIALS AND METHODS:

Line 105. What is meant by “eggs with morphology compatible for Ae. albopictus”? The language makes it sound like eggs were identified to species-level. I would suggest using “eggs with morphology compatible for container-breeding Aedes mosquitoes” for clarity.

AU: Thank you. The sentence has been modified according to the comments and suggestions of the reviewer (lines 115-116).

Lines 113-118. I appreciate the fact that mosquito control was carried out by a variety of independent operators who use different methods, but without knowledge of a) what formulation insecticide(s) were used, b) what quantity was used, and c) where applications were made, there is no way to determine the thoroughness or quality of the insecticide applications. Therefore, there is no way to tell if the treatments had an effect or not (maybe some treatments were very light, maybe some used an inappropriate chemical, maybe some did adulticide and some did not). There is no way to tell if mosquito control has an effect on Aedes albopictus numbers or not based on this information.

AU: No indeed, conclusions cannot be drawn about it. In the text we have tried to make this clearer.

Lines 120-126. More detail is needed on these bite cases. Does the patient bring the insect or a description of the insect when reporting the bite? How would the doctor know if it’s a bedbug, simuliid, chemical burn, environmental irritatant, or other medical condition? Oftentimes people report being bitten by arthropods when there is no evidence of arthropods. Without more detail (or perhaps evidence that the vast majority of bite reports are Aedes albopictus bites, at which point you can use total bite reports as a proxy for Aedes albopictus bites), you can’t really link changes in bite reports to presence of Aedes albopictus in an area.

AU: Occasionally the patient brings the insect to the Health Center. Moreover, we constantly receive ticks from different hospitals of the Basque Country in order to identify the species. Patients attend health services after an insect bite when the bite shows signs of irritation, inflammation or they suffer from an allergic reaction. Bite type is defined by the doctor who attends the person bitten, using the ICD-9 (International Classification of Diseases, 9th revision). The text has been modified (lines 139-141).

We agree with the reviewer and we cannot attribute the increase in bites in Bidasoa to Ae.albopictus, we have tried to indicate this on the text.

Lines 137-139. 95% confidence intervals were generated using SPSS, but was there any sort of statistical test done? That should be described here.

AU: In order to make the comparison of the bites rates by Health County and year, the calculation of the biting incidents per 1,000 inhabitants, with confidence intervals at 95%, was analysed by the SPSS software, using the formula: , where p is the incidence, zα/2 is 1.96 for 95% of confidence interval, and N is the sample size.

Figure 1. It would be helpful if the regions were labelled. It is not easy to cross-reference the figure with the text.

AU: Figure 1 has been modified according to the comments and suggestions of the reviewer. In addition, the size of the Basque Country map has been increased, in order to have a better identification of the three Basque Provinces, as they are cited several times on the text. Also, the size of the HC legend has been amplified, and the colours and initials of each HC can now be better identified. .

RESULTS

Line 151. “All the ovitraps examined in 2013 were negative for Ae. albopictus eggs”. According to Figure 1 and the Figure 1 legend, all of the ovitraps were positive. Could it be the legend has switched which ovitraps are positive and which are negative?

AU: Yes, we apologize for this error; the legend in Figure 1 is wrong, red colour represents the positive sites, whereas white represents the negative sites. In the new version of the manuscript this mistake has been amended.

Line 200. I assume you mean Bacillus thuringiensis israelensis? There are multiple strains of Bt, and israelensis is the most likely to be used against Ae. albopictus larvae in the habitats described in this manuscript.

AU: Yes indeed, the strain used is Bacillus thuringiensis israelensis, in the new version of the manuscript it has been modified (line 231).

Line 201. Diflubenzuron should not be capitalised.

AU: Thank you for your comment, in the new version of the manuscript is has been changed (line 232).

Lines 203-205. The method by which treatment was statistically assessed needs to be described more in Methods. What was the control?

AU: The efficacy of the applied treatments in terms of reduction of positive ovitraps has not been studied, although the evolution of the mean of egg per positive trap per year was analysed in the sampling sites where Ae.albopictus eggs were detected for two or more consecutive years, as an indirect measure of the effect of treatments. The objective was to compare if there had been a significant increase or decrease in the mean numbers of eggs per positive ovitrap over the years, or if they have remained stable. No significant increase was observed throughout the study period in any of the studied sites. Unfortunately no negative control was used. In the new version of the manuscript a comment has been added in order to make it clearer (lines 225-238).

DISCUSSION

Lines 230-269. This paragraph is too long. It should be broken down into three to five paragraphs.

AU: The paragraph has been modified according the suggestion of the reviewer.

Lines 243-244. I’m not sure what is meant by “short time detection”. Rapid detection?

AU: We are making reference to a sporadic detection of eggs in a particular sampling, and that does not repeat again. We have changed “short time detection” for “sporadic detection” (line 290).

Line 246. Replace “egg lying” with “egg laying”

AU: Change done. We apologize for this mistake (line 293).

Lines 256-258. “This temperature threshold…could explain the absence of adults during the first five years after the first detection of eggs”. I recommend replacing “absence of adults” with “failure to detect adults”. Clearly adults were not absent from the area, as they were laying eggs.

AU: Change done (line 305). Thank you for the clarification.

Lines 262-264. “However, it is difficult to confirm that Ae. albopictus was the solely reason for the increase in bites in 2018 since an increase in the population of native Culicidae has also been reported in several European countries due to the high temperatures in summer”. I think this really sheds doubt on the ability of increased biting reports, even in Bidasoa, to incriminate Ae. albopictus as the cause of the bites. Perhaps one message drawn from this manuscript should be the inclusion of biting surveys as part of Aedes albopictus surveillance, where mosquitoes are identified.  Also, change "solely" to "sole".

AU: Yes, we agree that drawing a message that human bite surveillance should be included in Aedes albopictus surveillance programs is interesting (lines 361-362).

We apologize for this mistake.

Lines 284-286. “In spite of these limitations, the average number of eggs per positive ovitraps and year did not increase over time in the sites with Ae. albopictus established populations (sites 7, 10, 20 and 23), suggesting that the applied biocide treatments helped to maintain low densities of mosquitoes”.   The results do not support this conclusion. An alternative hyposthesis is that Aedes albopictus populations grew so slowly over those years that their growth was undetectable. In order to show that insecticide treatment kept populations low, there would need to be consistency in insecticide applications (the type, amount and location of application would need to be stated, so the reader would know that a similar amount of control was applied in all areas), and there should be a control area where Aedes albopictus was untreated, or at least a comparison made with an area where Aedes albopictus were not treated, to show that Aedes albopictus populations will increase if untreated.

AU: The sentence has been suppressed. A new line according the comment and suggestion of the reviewer has been added (lines 306-307). We agree that in order to draw conclusions on the efficacy of the treatments applied in an area, a study should be specially designed and there should be a control negative area.

CONCLUSIONS

As stated before, I would consider ways to improve the correlation between Aedes albopictus numbers and reports of humans being bitten. One suggestion is to have human landing catches as part of the surveillance program, in order to measure biting rates accurately.

AU: Thank you for your comments. We will consider in the future using human landing catches as part of the surveillance programme. We have added it in the conclusions (line 361-362).

Thank you for your review and for all the comments, which have helped improve the paper.

Reviewer 4 Report

In the paper entitled “Progressive invasion of Aedes albopictus in Northern Spain in the period 2013-2018 and its impact on human bites”, Goiri et al. used mosquito surveillance to map the geographical distribution of Ae. albopictus in various counties of the Spanish Basque Country and assess its expansion throughout the years between 2013 and 2018. This invasive mosquito species is of particular concern due to the numerous diseases it can transmit to humans including Zika, dengue and chikungunya to name but a few. The topic is thus extremely interesting and relevant, and surveillance is essential for deploying efficient control strategies. Nevertheless, I think this paper is lacking important details, most importantly in the methods and results sections. The lack of consistency in sampling across the time period (2013-2018) makes it difficult for the reader to really dissociate the mosquito species expansion from the absence of sampling in some of the years across this period.

Specific comments are below:

Abstract: Please describe the type of method you use for collecting mosquitoes.

Introduction:

Please provide more backgrounds on Aedes albopictus biology (e.g., high degree of anthropophagy). The impact of climate change and the mechanisms underlying this species incredible ability to adapt to new habitats should be emphasize.

L49: using “rate of bites in people” does not appear as an accurate method to quantify the Ae. albopictus mosquito burden on local populations (which is by the way reflected by the results in Figure 4). Bites could be associated with other mosquito species (or other closely related blood sucking arthropods).  

Methods:

L99: “entomological surveillance took place between September and November” … For how long these traps were deployed? How many replicates are there per site? Please provide more details to this section.

L108: How many traps per site? For how long were they deployed? Were all the traps described setup at the same time? This part needs more technical details.

L113: This section also lacks details regarding treatments applied, frequency of application and most importantly what impact it had on subsequent mosquito sampling by the authors.

L136: Please provide the citation for the R software.

Results:

Where are the BG sentinel and prokopack data? How many total adult mosquitoes were trapped?

What about climatic conditions at the sampling sites? Were temperature and humidity recorded? These two factors are greatly affecting mosquito population dynamics and could be great to include to the study if available.

L194: I am curious about data collection pre / post treatment in terms of time frame and results.

L207: Looks like the “insect bite consultation” for 2014 is slightly higher for most of the location here.  

Discussion and Conclusion:

Elaborate on the limitations of the study.

Elaborate on the use of this type of data for mosquito population dynamic projections through mathematical modelling and highlight the potential impact of climate change on albopictus distribution in Europe.

Throughout the manuscript:

“mosquitos” should be spelled “mosquitoes”.

Tables S1 and S2 are not cited in the main manuscript.

Author Response

Reviewer 4

In the paper entitled “Progressive invasion of Aedes albopictus in Northern Spain in the period 2013-2018 and its impact on human bites”, Goiri et al. used mosquito surveillance to map the geographical distribution of Ae. albopictus in various counties of the Spanish Basque Country and assess its expansion throughout the years between 2013 and 2018. This invasive mosquito species is of particular concern due to the numerous diseases it can transmit to humans including Zika, dengue and chikungunya to name but a few. The topic is thus extremely interesting and relevant, and surveillance is essential for deploying efficient control strategies. Nevertheless, I think this paper is lacking important details, most importantly in the methods and results sections. The lack of consistency in sampling across the time period (2013-2018) makes it difficult for the reader to really dissociate the mosquito species expansion from the absence of sampling in some of the years across this period.

 Specific comments are below:

Abstract: Please describe the type of method you use for collecting mosquitoes.

AU: The methods used for collecting mosquitoes have been added in the abstract (line 25).

Introduction:

Please provide more backgrounds on Aedes albopictus biology (e.g., high degree of anthropophagy). The impact of climate change and the mechanisms underlying this species incredible ability to adapt to new habitats should be emphasize.

AU: A new sentence has been added (line 40-43) according to the suggestion of the reviewer.

L49: using “rate of bites in people” does not appear as an accurate method to quantify the Ae. albopictus mosquito burden on local populations (which is by the way reflected by the results in Figure 4). Bites could be associated with other mosquito species (or other closely related blood sucking arthropods).  

AU: The reviewer is right in all he/she says. The rate of insect bites in humans is only an indicator of the possible presence of the tiger mosquito in the area. In fact, in Catalonia (Spain), in 2004, the presence of tiger mosquito was detected due to the alarming increase in the rate of insect bites in humans through the years (Gimenez et al., 2007). In this study, we have tried to find out if the increase in Ae.albopictus populations in an area was related to the increase in the rate of bites by insects in the population, considering that it is impossible to verify which species has caused the bite. We have tried to indicate this in the new tittle and in the text (lines 67-69).

Methods:

L99: “entomological surveillance took place between September and November” … For how long these traps were deployed? How many replicates are there per site? Please provide more details to this section.

AU: At the beginning of the surveillance programme (2013-2015), the samplings were started in September and ended mid-November. Later on, the samplings were moved to July (2016-2018) and they ended in November. In each sampling site, 8-10 ovitraps were placed and these were maintained in the same location until the end of the sampling season (November). The traps were distanced from each other 15-20 meters and were placed in shaded locations and protected from the wind. Ovitrap sticks were replaced and transported to the laboratory weekly. As for the water of the trap, it was examined to see if there were any mosquito larvae, and, if present, were also transported to the laboratory for identification. All this is summarised in lines 102-116.

L108: How many traps per site? For how long were they deployed? Were all the traps described setup at the same time? This part needs more technical details.

AU: For adult trapping of mosquitoes, during 2015-2017, only 1-2 BG-Sentinel traps were available per site and a weekly sampling was made between September-November. In 2018, 2 prokopack aspirators and 5 gravid traps were implemented. In Material and Methods, the methodology is now better specified (lines 119-124). Besides, Table S3 has been included, describing the samplings made and the different species found.

L113: This section also lacks details regarding treatments applied, frequency of application and most importantly what impact it had on subsequent mosquito sampling by the authors.

AU: Unfortunately, we only have information for the type of active compound used and the dates when the treatment was applied. Not much emphasis has been placed on this part of the work, as it can only be descriptive and we cannot draw conclusions about it.

L136: Please provide the citation for the R software.

AU: The information has been added (line 157).

Results:

Where are the BG sentinel and prokopack data? How many total adult mosquitoes were trapped?

AU: A summary table has been included in the manuscript detailing the adult samplings performed, as well as, the obtained results (Table S3). This table had not been included before in the manuscript due to the low rate of captures obtained.

What about climatic conditions at the sampling sites? Were temperature and humidity recorded? These two factors are greatly affecting mosquito population dynamics and could be great to include to the study if available.

AU: We have compiled weekly meteorological data (min, max, mean temperatures, humidity and precipitation) from May to October, from the meteorological station located near the sampling site 10, where the highest Ae. albopictus population has been observed. We performed a quick and simple correlation analysis between the percentage of positive ovitraps and the number of eggs per trap and week, with the climatic variables mentioned above. No significant correlations were found between abundance of ovitraps / density of eggs, and meteorological variables. As we consider that deeper analyses are needed, we have not include these preliminary results in the manuscript

L194: I am curious about data collection pre / post treatment in terms of time frame and results.

AU: As treatments information is scarce and no negative control was included, we have decided to suppress the sentence since no analyses could be done.

L207: Looks like the “insect bite consultation” for 2014 is slightly higher for most of the location here.  

AU: The reviewer is right in all he/she says, but the only differences observed were in 2018 in the Health County of Bidasoa (B). Insect bites consultations in 2014 were slightly higher but not statistically significant.

Discussion and Conclusion:

Elaborate on the limitations of the study.

AU: A paragraph has been added in the discussion mentioning the limitations of the study (lines 275-280).

Elaborate on the use of this type of data for mosquito population dynamic projections through mathematical modelling and highlight the potential impact of climate change on albopictus distribution in Europe.

AU: A sentence was added with the idea proposed by the reviewer, and that we found interesting to add to the discussion (lines 344-346).

Throughout the manuscript:

“mosquitos” should be spelled “mosquitoes”.

AU: We apologize for this mistake; change has been done throughout the manuscript.

Tables S1 and S2 are not cited in the main manuscript.

AU: In the new version of the manuscript, Table S1 and Table S2 are mentioned in the corresponding section of the manuscript. And we have added two new tables (Table S3 and Table S4) also cited in the manuscript.

Thank you for your review and for all the comments, which have helped improve the paper.

Round 2

Reviewer 3 Report

All changes have been made satisfactorily.

One misspelling I missed in the previous draft was "S-methoprene" (line 260).

Reviewer 4 Report

The authors have addressed this reviewers' suggestions and comments.